health and disease and epidemiology/theoretical biology/computational biology

epidemic, class size, susceptible–exposed–infected–recovered, Gillespie

**Author for correspondence:**
Alex Best
e-mail: a.best@shef.ac.uk

# The impact of varying class sizes on epidemic spread in a university population

Alex Best, Prerna Singh, Charlotte Ward, Caterina Vitale, Megan Oliver, Laminu Idris and Alison Poulston

School of Mathematics and Statistics, University of Sheffield, Sheffield S3 7RH, UK

 AB, 0000-0001-6260-6516

A common non-pharmaceutical intervention (NPI) during the COVID-19 pandemic has been group size limits. Furthermore, educational settings of schools and universities have either fully closed or reduced their class sizes. As countries begin to reopen classrooms, a key question will be how large classes can be while still preventing local outbreaks of disease. Here, we develop and analyse a simple, stochastic epidemiological model where individuals (considered as students) live in fixed households and are assigned to a fixed class for daily lessons. We compare key measures of the epidemic—the peak infected, the total infected by day 180 and the calculated $R_0$—as the size of class is varied. We find that class sizes of 10 could largely restrict outbreaks and often had overlapping inter-quartile ranges with our most cautious case of classes of five. However, class sizes of 30 or more often result in large epidemics. Reducing the class size from 40 to 10 can reduce $R_0$ by over 30%, as well as significantly reducing the numbers infected. Intermediate class sizes show considerable variation, with the total infected varying by as much as from 10% to 80% for the same class size. We show that additional in-class NPIs can limit the epidemic still further, but that reducing class sizes appears to have a larger effect on the epidemic. We do not specifically tailor our model for COVID-19, but our results stress the importance of small class sizes for preventing large outbreaks of infectious disease.

# 1. Introduction

The classic susceptible–infected–recovered (SIR) epidemiological model has long been used to model the spread of infectious disease in human, animal and plant populations [1,2]. More recently, its extended susceptible–exposed–infected–recovered (SEIR) framework has formed a central pillar of much of the modelling of the COVID-19 pandemic, often including highly realistic movement and contact networks [3–6]. A key non-pharamceutical intervention (NPI) for populations across the world during the COVID-19

pandemic has been restricting population mixing through 'lockdowns', with people encouraged to stay at home and avoid mixing with individuals outside their household unless essential. This has often included closing educational settings of universities and schools, with 31 countries enacting full school closures and reduced schooling in a further 48 countries [7]. As countries move to reopen these settings, an important question is how classes can be organized to minimize further disruption to students' education while limiting epidemic spread. There have been some excellent, in-depth modelling studies of infection spread in educational settings, especially universities, with a range of NPIs included, often with a focus on testing and isolation strategies [8–11]. Here, we focus on the question of how class sizes may impact an epidemic.

An important measure of infectious disease growth and severity is the basic reproductive ratio, $R_0$ [12,13]. This well-known term defines the average number of new infections from one infected individual in an otherwise disease-free population. In mathematical definitions, $R_0$ is broadly the product of three quantities: the potential (or probability) of infection upon an infectious contact, the infectious period and the number of disease-free individuals contacted per unit time [14]. An important consequence of this is that the larger the population that can be contacted by an infectious individual—the effective population size—the greater the potential spread of the epidemic. Thus, if a population can be partitioned into smaller sub-groups with minimal mixing between them, the spread of disease can be significantly limited. This is a reality populations all over the world have experienced during the COVID-19 pandemic, with 'lockdown' measures aimed at limiting mixing of households or groups. Data suggest that such NPIs—such as closing businesses, closing schools and, of relevance to our study, limiting group gathering sizes—have reduced the $R_0$ of COVID-19 by as much as 60% [15]. While there are many studies examining epidemic spread in heterogeneous environments and/or on networks, to our knowledge few modelling studies have specifically explored how group size limits might impact the extent and severity of an epidemic. Kain *et al.* [5] found that 'chopping off the tail' of individual infection distributions—in effect preventing large gatherings—could effectively restrict an epidemic in their COVID-19 parametrized model, reducing both mean transmission and the variance in outcomes. In their broader examination of COVID dynamics on a university campus, Brook *et al.* [9] found that a group size limit of six could reduce the effective reproductive ratio from 1.05 to 0.86, but limits of 50 had almost no impact on disease spread. As we seek to reopen universities and schools, some mixing between households will be essential. An important consideration, then, is the degree to which we could allow some mixing while still limiting the extent of the epidemic.

Ultimately, our study investigates what happens to an epidemic when a population partitioned into households is mixed for a short time period each day into fixed groups. The situation loosely in our minds is of a university cohort living in accommodation who attend a class each day. Similarly, we might consider a school population attending classes, or a local community forming interaction bubbles. However, we stress that our study is a relatively simple, theoretical study of the impact of mixing, and we make no strong claims about the precise values or predictions our model makes. For clarity and transparency, we state here a few of our key assumptions: (i) all classes occur simultaneously with immediate transitions between class and home, (ii) we do not include testing and isolation, (iii) there is no further mixing than in classes and homes, either within the population or externally, and (iv) we do not include 'superspreaders' but instead assume all individuals experience the same transmission rates. We discuss the possible impact of each of these assumptions in the Discussion. We also do not attempt to parametrize or structure our model specifically for COVID-19. Rather, we seek to identify the general patterns that result from mixing partitioned populations into different sized groups.

## 2. Methods

We develop and run stochastic simulations of an epidemiological model using python (code is available from GitHub, https://github.com/abestshef/classsizeSEIR). The underlying epidemiological model is an SEIR framework where, within a setting (home or class), the dynamics would be given by the following ordinary differential equations:

$$\frac{dS}{dt} = -\beta SI, \tag{2.1}$$

$$\frac{dE}{dt} = \beta SI - \omega E, \tag{2.2}$$

$$\frac{dI}{dt} = \omega E - \gamma I \tag{2.3}$$

and

$$\frac{dR}{dt} = \gamma I, \tag{2.4}$$

where $\beta$ is the transmission coefficient (with $\beta I$ the 'force of infection'), $\omega$ is the rate of progression from exposed to fully infected and $\gamma$ is the recovery rate. We note that our state variables are defined as densities, not proportions, and that the total population size is $N = S + E + I + R$ rather than being normalized to 1. The stochastic simulations use a Gillespie algorithm [16] to calculate waiting times between events. The possible events are initial infection $(S \rightarrow E)$, progression to full infection $(E \rightarrow I)$ and recovery $(I \rightarrow R)$. Which event occurs at a chosen time point depends on their relative probabilities at that point and in the relevant setting. Each day is divided into fixed time periods where all students are in each setting, either home or class. We assume immediate movement between settings, with classes occuring during $t \in [\text{day} + 0.4, \text{day} + 0.5]$, roughly equivalent to a 2.5 h period. The event probabilities will be different in each setting; thus when a transition time is reached, the waiting time is stopped and recalculated from the transition point. Besides these transitions, infection is assumed to occur evenly throughout the 24 h period.

The total population size is $N = 1000$ and students are randomly divided into houses of size $n_h$ and classes of (target) size $n_c$. We take two household sizes, $n_h = 10$ (100 households) and $n_h = 5$ (200 households; arguably reasonable averages for university halls and private housing respectively). Every household has exactly $n_h$ individuals. Target class sizes vary in steps of five from five to 50. Where $N/n_c$ is an integer, exactly that many classes are created, all containing exactly $n_c$ individuals. Where this is not the case, the classes contain either $n_c$ or $n_c - 1$ individuals (for example, for a target class size of 15, there are 62 classes of 15 students and five classes of 14 students). Both the house and class composition is fixed in each simulation. We randomly choose 25 individuals to be infected at the beginning of each simulation, and all other individuals are susceptible (we do not investigate the effect of varying this initial number). Our key investigation will be to vary average class sizes and explore the impact on the epidemic. We also compare results where transmission is high $(\beta = 0.5\gamma)$ and low $(\beta = 0.2\gamma)$. These values are chosen partly to give reasonably realistic estimates for $R_0$ and also because the lower value represents a 60% reduction of the higher value, which we use to represent a reduction due to NPIs (see below). While these values would appear to produce very high values of the basic reproductive ratio, $R_0$, in the mean-field model $(\beta = 0.5\gamma, N = 1000 \Rightarrow R_0 = \beta N/\gamma = 500)$, it is well known that the actual $R_0$ is considerably lower in individual-based models, especially when interactions networks are small [17]. We directly calculate $R_0$ from our simulations (see below) and found across all the results presented here the median $R_0$ fell in the range [0.66,3.68]. We also additionally examine the case where NPIs in the class (e.g. masks, ventilation, distancing) reduce transmission from the high to the low value (a reduction of 60%). We additionally assume $\gamma = 1/14$ and $\omega = 1/7$ in all simulations, giving a latent period of 7 days and infectious period of 14 days.

Recent work has highlighted the difficulties in representing outcomes from stochastic epidemic models [18]. First, to visualize the 'typical' time courses, we follow the methods of Juul *et al.* [18] to present the 'most central' 50% of simulation runs. One hundred simulations are run, discretized and stored. We then repeatedly sample subsets of these stored runs (100 samples of 20 curves) and increase the 'score' of any run that falls entirely within the bounds of the sampled curves between time-points 10 and 150. Secondly, we present three key measures of the epidemic—the peak number infected, the total number infected by day 180 and the calculated $R_0$ (see below)—from 100 simulation runs for each class size using box and whisker plots. These highlight the median values, the inter-quartile range (IQR; 25%–75%), the maximum/minimum (or $1.5 \times$ IQR if smaller) and any outliers (values greater than $1.5 \times$ IQR). Alongside these, we compare the IQR of the class of five (the 'most cautious' approach) with all other class sizes, noting where the IQRs do and do not overlap using shading of the boxplots. This allows us to explore whether class sizes can be raised above this cautious level without causing large changes to the outcome.

A brief note on the basic reproductive ratio, $R_0$; in our simple SEIR structure, we would have $R_0 = \beta \hat{N}/\gamma$, which depends on the effective disease-free population size $\hat{N}$. However, the interpretation of $\hat{N}$ will vary depending on the degree of mixing. Here, we make a direct calculation of $R_0$ in each simulation by recording the number of infections caused by the 25 initially infected individuals, an intuitive measure of $R_0$ as might be estimated during a real epidemic.

# 3. Results

## 3.1. Large households

Taking an average household size of 10 and comparing the most central 50% of runs for average class sizes of 10 and 40 (figure 1*a*–*c*), it is clear that smaller class sizes substantially restrict the epidemic.

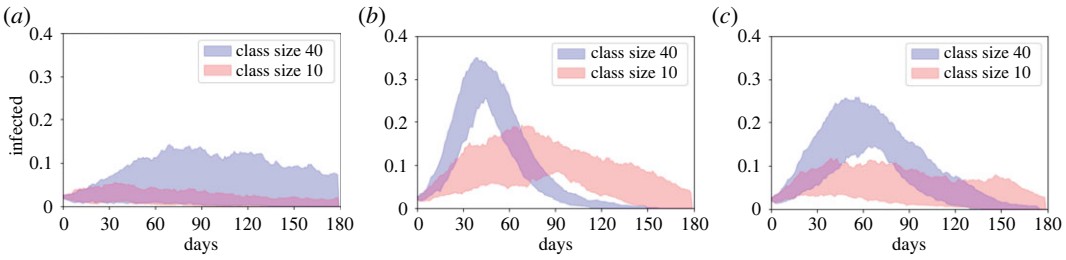

**Figure 1.** Bounds of the 'most central' 50% of 100 simulation runs for large average households for average class sizes of 40 (blue) and 10 (red) where infection rates are (a) low, $\beta = 0.2\gamma$, (b) high, $\beta = 0.5\gamma$ and (c) high at home but low in class.

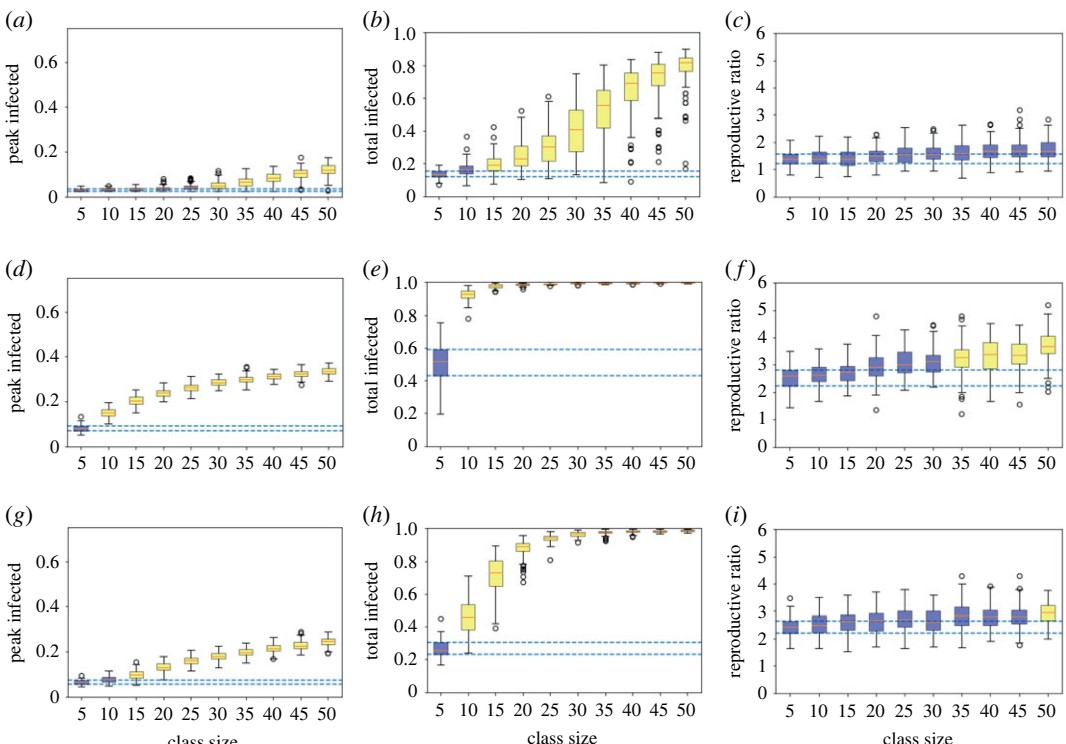

**Figure 2.** Measures of epidemic severity for household size of 10 for different (target) class sizes from 100 simulation runs at each class size. Boxplots of the peak (a,d,g) and total infected by day 180 (b,e,h) and the calculated $R_0$ (c,f,i), where infection is low, $\beta = 0.2\gamma$ (a–c), high, $\beta = 0.5\gamma$ (d–f) and high at home but low in class (g–i). The orange lines denote the median, the boxes the 25th and 75th centiles, the whiskers to 1.5 the interquartile range and circles any outliers. The dashed lines mark the inter-quartile range (IQR) for the class of five, and colouring of the boxes whether the IQRs of each class size do (blue) or do not (yellow) overlap the class of five's IQR.

When infection rates are low ($\beta = 0.2\gamma$, figure 1a), with a class size of 40 there is considerable variability, with some of the central curves showing minimal spread but others reaching peaks above 15% infected. Reducing the class size to 10 clearly restricts the central epidemics, with few curves peaking above 5% infected and in some cases the epidemic completely finishing by day 60. For greater infection rates ($\beta = 0.5\gamma$, figure 1b), there is a clear epidemic in all of the central runs for any class size, but is clearly more severe with the larger groups, with the peak of the central runs increasing from never more than 20% for a class size of 10 to always more than 26% for a class size of 40. Finally, we investigate the impact of having simple NPIs in place in classes such that the infection is reduced (from $\beta = 0.5\gamma$ to $\beta = 0.2\gamma$) while in class but not at home. Compared to the previous case we do see reductions in the epidemic, with the peaks lowered by around 10%. Noticeably, however, solely reducing the class size from 40 to 10 (figure 1b blue versus red) causes a greater reduction in the epidemic than solely instituting the in-class NPIs in a class of 40 (figure 1b blue versus figure 1c blue).

Looking in more detail for varying class sizes using the boxplots, with low infection rates (figure 2a–c), we again clearly see that greater class sizes lead to larger epidemics in terms of all three

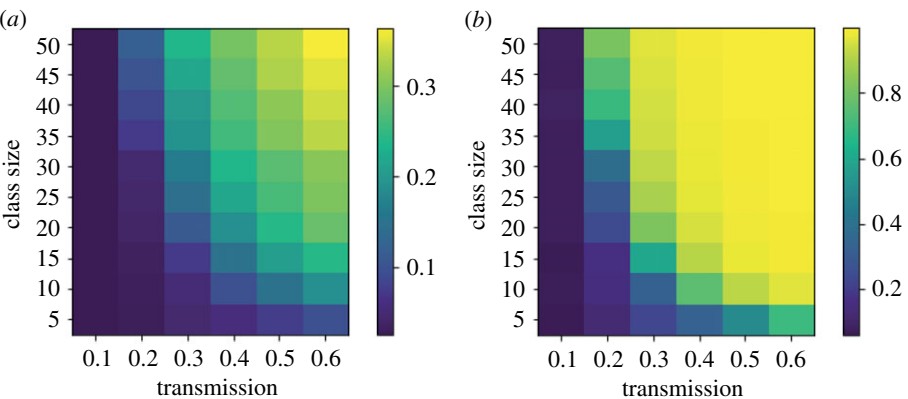

**Figure 3.** Heatmaps of median peak and total infections for varying (target) class sizes and transmission (where $\beta$ is the figure given multiplied by $\gamma$. The household size was 10 and there were 50 simulation runs for each combination of class size and transmission rate.

measures. The colouring highlights that only a class size of 10 has an overlapping IQR for both peak and total infections, while class sizes up to 25 have an overlapping IQR for peak infections only, meaning sizes of 30 or above have clearly different outcomes to a class of five. Moreover, for a class size of 10 the top of the inter-quartile range (IQR) is 19% total infecteds, but for a class size of 40 the bottom of the IQR is 58%, emphasizing the large effect of different sizes. While all class sizes have overlapping $R_0$ IQRs with the class of five, since this is essentially a logarithmic quantity of epidemic growth it is not unexpected, and the median value is reduced from 1.68 for a class size of 40 to 1.40 for a class size of 10.

We see considerable variation for intermediate class sizes, with the minimum and maximum total infected for classes of 35 and 40 extending from below 10% to above 80%, suggesting different locations could experience very different epidemics purely due to stochastic variation. We also note that we assumed the same number of initial infections in all simulation runs. Different institutions would probably start with different number of infections, which would further increase the heterogeneity of outcomes.

When infection rates are larger (figure 2$d$–$f$), there are very large epidemics no matter the class size, especially in terms of the total number infected. There are clearly no class sizes where the IQRs overlap with the smallest class for the peak and total infected, while only class sizes of 30 or smaller have overlapping IQRs for $R_0$. A class size of 15 or above results in more than 94% of the population infected in every single simulation run. Smaller class sizes do lead to noticeably lower peaks, however—the top of the IQR for class sizes of 15 is 22% and for a class size of 10 it falls to 17%. Reducing the class size from 40 to 10 also leads to a drop in the median $R_0$ from 3.40 to 2.64 (a 22% reduction).

In-class NPIs lead to a modest reduction in the severity of the epidemic at all class sizes (figure 2$g$–$i$), though the epidemic remains significantly larger for larger class sizes. Interestingly, a class size of 10 with no NPIs (peak IQR 14%–17%, total IQR 91%–95%) generally results in smaller epidemics than a class size of 40 with NPIs (peak IQR 21%–23%, total IQR 97%–99%). Thus group size limits in themselves may lead to better outcomes than many other mitigation measures (given our assumptions). The combination of small class sizes and the in-class NPIs can dramatically reduce the severity of the epidemic. Comparing a class size of 40 without NPIs to a class size of 10 with NPIs, the median peak is reduced from 32% to 8%, the median total from 99% to 64% and the median $R_0$ from 3.40 to 2.48. The class size of 10 has an overlapping IQR with the class of five for the peak infected, but no classes overlap for total infected.

For completeness, we also ran simulations with a finer resolution of transmission coefficient, $\beta$, and recorded the median peak and total infected (figure 3). At the lowest transmission ($\beta = 0.1\gamma$), there are no outbreaks for any class size. However, as the transmission increases we clearly see the increasing important of smaller class sizes, particularly in terms of total numbers infected.

## 3.2. Small households

When the average household is reduced to five, comparing figures 1 and 4 shows that the size of the epidemic is reduced in all cases, since there is naturally less mixing between individuals. When

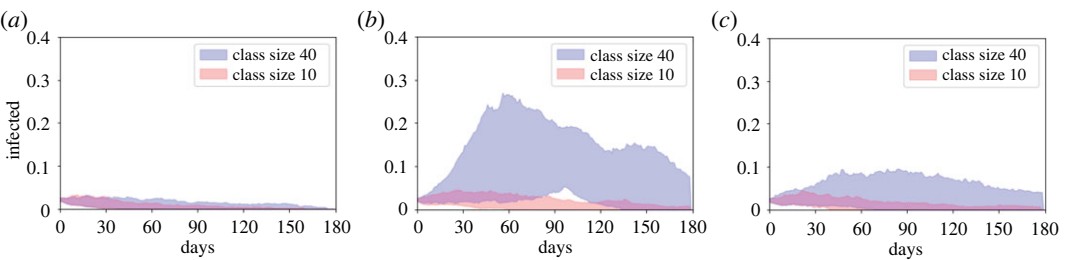

**Figure 4.** Bounds of the 'most central' 50% of 100 simulation runs for small average households for average class sizes of 40 (blue) and 10 (red) where infection rates are (a) low, $\beta = 0.2\gamma$, (b) high, $\beta = 0.5\gamma$ and (c) high at home but low in class.

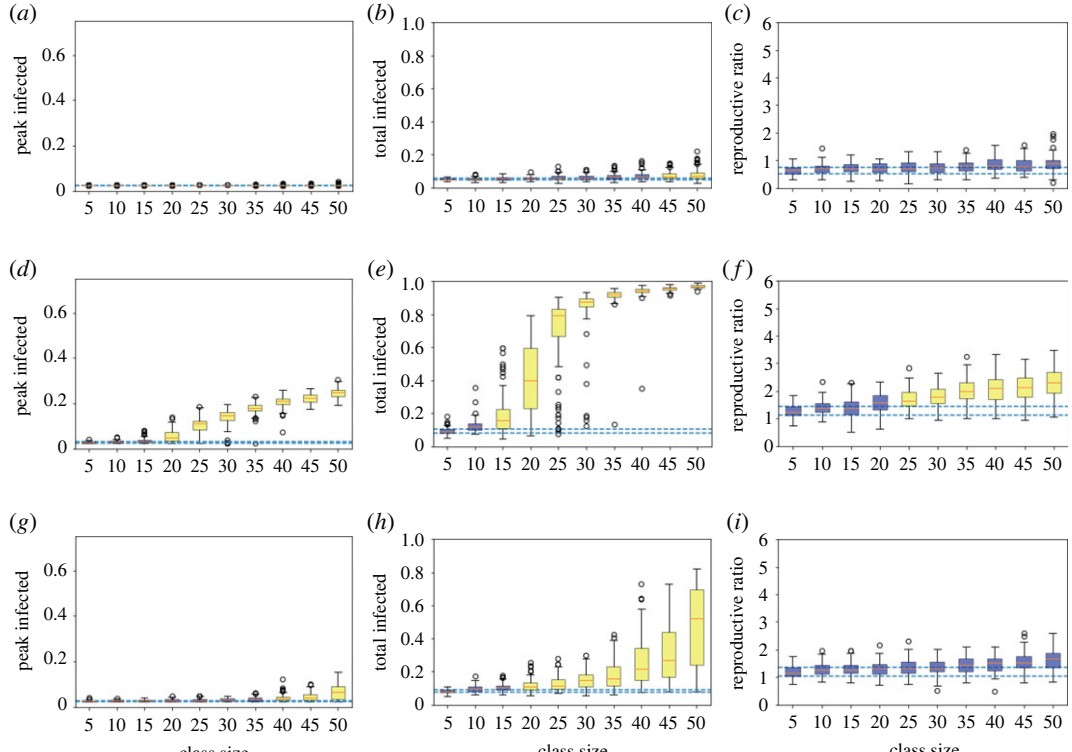

**Figure 5.** Measures of epidemic severity for household size of five for different (target) class sizes from 100 simulation runs at each class size. Boxplots of (a,d,g) the peak and (b,e,h) total infected by day 180 and (c,f,i) the calculated $R_0$, where infection is (a–c) low, $\beta = 0.2\gamma$, (d–f) high, $\beta = 0.5\gamma$ and (g–i) high at home but low in class. The orange lines denote the median, the boxes the 25th and 75th centiles, the whiskers to 1.5 the interquartile range and circles any outliers. The dashed lines mark the inter-quartile range (IQR) for the class of five, and colouring whether the IQRs of each class size do (blue) or do not (yellow) overlap the class of five's IQR.

transmission is low ($\beta = 0.2\gamma$, figures 4a and 5a–c) there are no significant outbreaks for any class size. For all class sizes considered the median $R_0$ is less than 1, and the peak is lower than 5% in every simulation run. All class sizes' peak IQRs overlap with the class of five's IQR, and classes of 40 or smaller have overlapping IQRs for total infected.

We see dramatic impacts of reducing the class size when transmission is higher ($\beta = 0.5\gamma$, figures 4b and 5d–f). Reducing the class size from 40 to 10 reduced the median $R_0$ from 2.12 to 1.40 (a 33% decrease). Even classes of 25 lead to significant outbreaks with the bottom of the IQR being 67% for the total and 8% for the peak, whereas for a class size of 10 the top of the IQR is 14% for total infected and 3% for the peak. Compared to the class of five, only classes of 10 and 15 have an overlapping IQR for the peak infected and only classes of 10 overlap for total infected. We again see considerable variation in outcomes for fixed class sizes, with total infected in class sizes of 20 and 25 stretching from a minimum of below 10% to a maximum of above 80%.

When NPIs are included in the class setting (figures 4c and 5g–i), for a class size of 40 the median peak is reduced from 21% without the NPI to 4% with it, and the median $R_0$ from 2.12 to 1.52. These

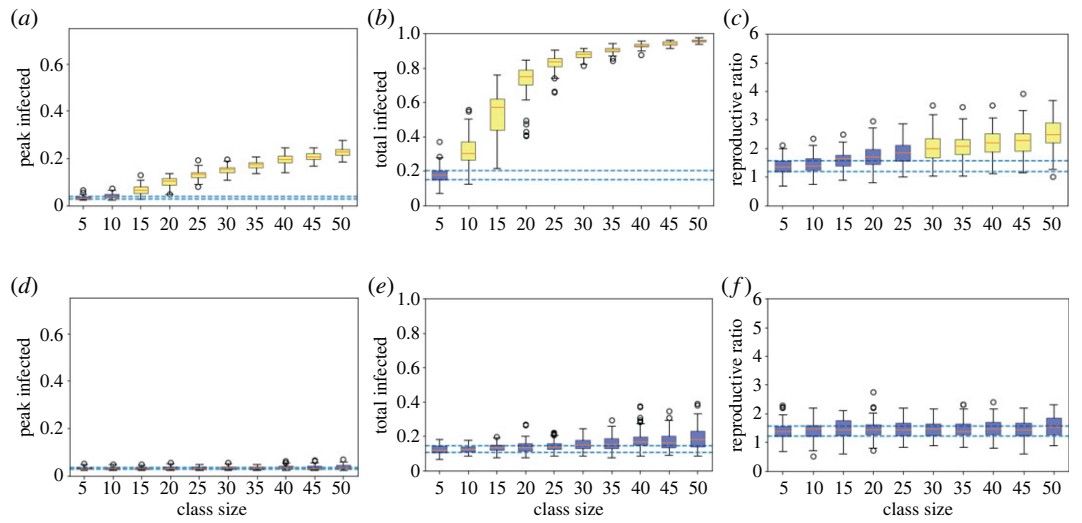

**Figure 6.** Box plots for peak infected, total infected and $R_0$ when the class length is doubled (*a–c*) or halved (*d–f*). $\omega = 1/7$, $\gamma = 1/14$. $\beta = 0.2\gamma$ in both settings. See earlier figures for description.

figures make it roughly equivalent to a class size of 15 without NPIs. We again see that the epidemic is more severe for a class size of 40 with NPIs than for a class size of 10 without (respective median peaks: 4% versus 3%, median totals: 22% versus 12%, median $R_0$: 1.52 versus 1.40). Combining both smaller classes and in-class NPIs can have substantial impacts: comparing a class size of 40 without NPIs to a class size of 10 with NPIs, we see the median peak reduced from 21% to 3%, the median total infected from 94% to 10% and median $R_0$ from 2.12 to 1.26. Compared to the class of five, class sizes of 35 and below have overlapping IQRs for the peak, but only class sizes of 10 and 15 for the total infected. We also see significant variation in outcomes for larger class sizes.

## 3.3. Time in class

We additionally consider what happens when we alter the time in class to be double ($t \in [\text{day} + 0.4, \text{day} + 0.6]$) or halved ($t \in [\text{day} + 0.4, \text{day} + 0.45]$) compared to above, assuming large households ($n_h = 100$) and low infection rates ($\beta = 0.2\gamma$). Predictably, increasing the class length leads to larger epidemics and decreasing it leads to smaller ones (figure 6). For the shorter classes, all class sizes peak and total IQRs overlap with the class of five. Interestingly, $R_0$ is found to be similar for any class size, the median varying only from 1.40 to 1.54. By contrast, no class sizes have overlapping IQRs for total infected and only classes of 10 for peak infected when the class length is doubled.

## 4. Discussion

As might be expected, a clear result from our model is that the smaller the class size, the lesser the severity of the epidemic. For our paramater sets, reducing the class size from 40 to 10 showed reductions in the median $R_0$ of up to 30%. Within that, our results suggest that 'optimal' class sizes across all measures of epidemic severity will rarely exist. Instead, decisions may vary depending on the aim, for example whether to ensure a low peak (to limit pressure on health services) or a low number of total infections (to protect as many individuals as possible from infection). Broadly speaking, small increases in class size from small groups initially lead to only modest increases in the peak of infections but rapid increases in total infections. For example, for high infection rates and small households, increasing the class size just from 10 to 15 led the median peak to increase from 5% to 9% but the median total infected from a modest 27% to a substantial 66%. The average household size and time in class also impacted the severity of outbreak, with larger households and longer classes predictably increasing the potential for large epidemics. In these cases, small class sizes were even more essential. Institutional decisions are therefore likely to depend on the desired outcome and specific local conditions, and transparency of decision making will be crucial.

If we were to trust our values here as being representative of a real university (which we would caution should be in the light of the many assumptions underlying the model), we would suggest

that to ensure the best chance of a restricted epidemic then classes should be limited to 10. In many cases, this would have an overlapping inter-quartile range with—and thus not be clearly different to—the most cautious approach of classes of five for the peak and total infected. Slightly larger classes—up to 25—may prevent the peak from rising too high, but will probably result in large numbers of total infections. We would strongly recommend against larger class sizes than 25 based on our parameters and assumptions as both peak and total infected were then consistently clearly different to the most cautious case of a class of five. In their more detailed study applied to a specific institution, Brook *et al.* [9] similarly found that small group size limits were the key NPI to reducing infection, showing that a limit of six could reduce the effective reproductive ratio from 1.05 to 0.86, but limits of 50 had almost no impact on disease spread. Similarly, Kain *et al.* [5] found that 'chopping off the tail' of individual transmission distributions—effectively preventing the grouping of large numbers of individuals—could be a key control measure. Moreover, based on data, Brauner *et al.* [15] found that restricting gatherings in all settings to 10 people or fewer was one of the most successful measures at reducing $R_0$ during COVID-19. It thus appears a consistent result that limiting group sizes to around 10 can be a successful NPI for slowing or even stopping epidemics.

The stochastic simulations reveal considerable variation in the epidemic time courses, particularly for intermediate class sizes. In some cases, the maximum and minimum of total infecteds in the 100 simulations spread from less than 20% to more than 80%. Thus while methods exist for approximating individual-based models with deterministic systems of ordinary differential equations [19–21], we highlight the importance of using stochastic simulations to appreciate the variety of possible outcomes. In practice, this demonstrates that institutions may make the same decisions about class sizes but experience very different epidemic time courses. As such, institutions may need to consider their 'risk appetite' for organizing logistically easier bigger class sizes at the risk of a large epidemic. Again, though, we emphasize that limiting class sizes to 10 or fewer largely prevented significant epidemics.

We investigated a simple case of employing NPIs in the class setting that would reduce the infection rate by 60%. We note that this would be a rather strong effect compared to estimates of NPI impacts from data [15,22]. This reduction was assumed to be due to simple NPIs such as social distancing but the exact method was not explicitly modelled. We found that, as would be expected, this led to smaller epidemics than if no NPIs were present, reducing $R_0$ by up to 25% for large class sizes. However, we consistently found that the epidemic was smaller in a class size of 10 without NPIs than a class size of 40 with NPIs. Given that the 60% reduction due to the NPI is already rather strong, this would suggest that reducing the class size may be the most efficient control measure. Both models [9] and data analysis [15] have similarly found that group size limitations was one of the most effective NPIs to prevent spread of COVID-19. Of course, both reducing the class size and implementing the NPI could reduce the epidemic considerably, in some cases reducing the median total infected from 96% to just 14% in our model.

As we have stated, we have relatively modest ambitions in this study of exploring how the size of mixing groups (considered as classes here) impact the time course of epidemics in partitioned populations. Other studies have provided highly detailed analyses of models with many realistic assumptions, contact networks and NPIs included [3–6] including in university settings [8–11]. If our study were to be applied to more realistic scenarios or to form the basis of decision making, some key further additions would be necessary. There are some key additions we would highlight. Firstly, we have assumed that all classes occur simultaneously and that transitions are immediate. In reality, classes would probably be spread throughout the day and there would be unavoidable mixing during transitions. Spreading the classes out would lower the effective size of the household population for much of the day, potentially reducing infection, while increased mixing during transitions would act oppositely. Secondly, we should consider additional NPIs, most importantly isolation of symptomatic (and possibly asymptomatic) individuals, as have been included in other models [8–11]. Given that any sort of isolation—whether due to infection or simply imposed—will reduce the degree of mixing and effective population size, such an approach would clearly be expected to further limit the epidemic. Also, we have assumed a closed population with no mixing outside of households or classes and full adherence by the population. We should account both for further external contacts and for additional mixing between individuals due to socializing, which we would expect to increase the potential for infection. We also assumed that all individuals have the same transmission rate when infected, but there is likely to be individual variation leading to 'super-spreaders' [23] and estimates from COVID-19 data suggest a high degree of super-spreading occurring [24]. This is likely to lead to greater variation in outcomes, with super-spreading often leading to rarer but larger outbreaks [23]. Furthermore, we assume that all individuals within each household attend exactly one class with the same cohort of

classmates. We do not account for the fact that (i) students may live in households with individuals who do not attend classes (family or other friends, for example) or (ii) that students may attend multiple classes. The latter in particular will lead to increased mixing of the population and thus increased transmission. While these additions would undoubtedly change the quantitative values found here, we would expect the fundamental findings—that smaller class sizes lead to smaller epidemics with less variation and that the patterns will vary according to the target measure—will remain.

Data accessibility. This article does not contain any additional data.
Competing interests. We declare we have no competing interests.
Funding. We received no funding for this study.
Acknowledgements. Many thanks to Alexander Fletcher for input while developing the stochastic simulation code.

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
