## [Peer Review File · Royal Society Open Science]

Review History

Decision letter (RSOS-210712.R0)

Dear Dr Best:

I am pleased to inform you that your manuscript entitled "The impact of varying class sizes on epidemic spread in a university population" is now accepted for publication in Royal Society Open Science.

If you have not already done so, please remember to make any data sets or code libraries 'live' prior to publication, and update any links as needed when you receive a proof to check - for

instance, from a private 'for review' URL to a publicly accessible 'for publication' URL. It is good practice to also add data sets, code and other digital materials to your reference list.

You can expect to receive a proof of your article in the near future. Please contact the editorial office (openscience@royalsociety.org) and the production office (openscience_proofs@royalsociety.org) to let us know if you are likely to be away from e-mail contact – if you are going to be away, please nominate a co-author (if available) to manage the proofing process, and ensure they are copied into your email to the journal. Due to rapid publication and an extremely tight schedule, if comments are not received, your paper may experience a delay in publication.

on behalf of Professor Joshua Ross (Associate Editor) and Professor Mark Chaplain (Subject Editor).

Associate Editor Professor Joshua Ross Comments to Author:
Comments to the Author:

Please ensure figures are legible at print size (i.e., fonts may need to be larger etc.), and that captions are free of typographical errors.
